# Advances in Roles of Salicylic Acid in Plant Tolerance Responses to Biotic and Abiotic Stresses

**DOI:** 10.3390/plants12193475

**Published:** 2023-10-04

**Authors:** Weiyi Song, Hongbo Shao, Aizhen Zheng, Longfei Zhao, Yajun Xu

**Affiliations:** 1School of Biology and Food, Shangqiu Normal University, Shangqiu 476000, China; songweiyi2008@126.com (W.S.); sqzaz@163.com (A.Z.); hnzhaolongfei@163.com (L.Z.); nmgxuyajun@163.com (Y.X.); 2Key Laboratory on Agricultural Microorganism Resources Development of Shangqiu, Shangqiu 476000, China; 3Jiangsu Key Laboratory for Bioresources of Saline Soils, Jiangsu Synthetic Innovation Center for Coastal Bio-Agriculture, Yancheng Teachers University, Yancheng 224002, China; 4Salt-Soil Agricultural Center, Institute of Agricultural Resources and Environment, Jiangsu Academy of Agriculture Sciences (JAAS), Nanjing 210014, China

**Keywords:** salicylic acid, biotic stresses, abiotic stresses, tolerance, immunity, yield

## Abstract

A multitude of biotic and abiotic stress factors do harm to plants by bringing about diseases and inhibiting normal growth and development. As a pivotal signaling molecule, salicylic acid (SA) plays crucial roles in plant tolerance responses to both biotic and abiotic stresses, thereby maintaining plant normal growth and improving yields under stress. In view of this, this paper mainly discusses the role of SA in both biotic and abiotic stresses of plants. SA regulates the expression of genes involved in defense signaling pathways, thus enhancing plant immunity. In addition, SA mitigates the negative effects of abiotic stresses, and acts as a signaling molecule to induce the expression of stress-responsive genes and the synthesis of stress-related proteins. In addition, SA also improves certain yield-related photosynthetic indexes, thereby enhancing crop yield under stress. On the other hand, SA acts with other signaling molecules, such as jasmonic acid (JA), auxin, ethylene (ETH), and so on, in regulating plant growth and improving tolerance under stress. This paper reviews recent advances in SA’s roles in plant stress tolerance, so as to provide theoretical references for further studies concerning the decryption of molecular mechanisms for SA’s roles and the improvement of crop management under stress.

## 1. Introduction

A wide range of stress factors in the plant growth environment, such as pathogen infection, drought, salt stress, etc., have a significant impact on plant survival and development [1,2,3,4]. In order to adapt to stress, plants have evolved a series of tolerance mechanisms involving the regulation of multiple cell signaling pathways [4,5,6,7]. Among them, salicylic acid (SA) plays an important role in plant stress tolerance [8,9,10]. This article will discuss the research progress of SA’s involvement in the cell signaling pathways of plants in combating stress.

It is generally held that SA plays vital roles in plant stress tolerance. SA’s main roles in plant tolerance to stresses can be summarized as following: the activation of plant defense mechanisms [2,11,12,13,14], the improvement of photosynthesis [15,16,17,18,19,20,21,22], the regulation of plant normal growth and development [8,19,23,24,25,26,27,28,29,30], and the enhancement of plant tolerance to drought [9,11,31,32,33,34,35,36], cold [37,38,39], heat [36,39,40,41,42], salt [43,44,45,46,47,48], and heavy metal stresses [49,50,51,52].

It was widely reported and reviewed that SA directly participates in the activation of plant defense mechanisms [2,11,12,13,14]. During its activation of relevant plant defense mechanism, SA can enhance the activities and expressions of antioxidant enzymes, including superoxide dismutase (SOD), catalase (CAT), and peroxidase (POD) [11,20,53,54,55,56,57,58,59]. Meanwhile, SA can induce the biosynthesis of active anti-pathogenic substances [1,60,61,62,63,64,65,66], thereby enhancing plant tolerance to biotic stresses caused by pathogens, fungi, and pests.

It was also generally reported and reviewed that SA signaling pathways act to increase chlorophyll contents in plant leaves and help to improve photosynthetic efficiencies [15,16,17], thereby significantly enhancing crop yield and other yield-related physiological indexes under various stresses [8,67,68,69].

It was also generally considered that SA can be involved in the regulation of normal growth and development of plants [8,19,23,24,25,26,27,28,29,30]. For example, SA can promote the differentiation of blossom buds, thus directly enhancing flowering [70,71]. On the other hand, SA can also improve number and quality of flowers. In addition, SA also exerts modulatory effects on seedlings by regulating seed dormancy and growth [72], thereby making them more vigorous or less susceptible to stresses [11,31,52,73,74,75,76]. New research has shown that SA plays multiple roles in enhancing plant stress resistances [32,38,43,73,77]. Such roles of SA can be categorized into the following aspects: immune response [13,25,78,79,80], antioxidative defense [11,20,53,54,55,56,57,58,59], salt tolerance [43,44,45,47,48,81,82,83,84], and drought tolerance [2,11,32,33,34,35,36,71,85,86,87].

A wide range of studies have found that SA is an important signaling molecule in the immune response of plants to biotic and abiotic stress [88]. It can activate the expression of a wide range of disease resistance genes [89], regulate cell wall strengthening [90], and promote hormone production to enhance plant tolerance to diseases [5,37,38,49,74,91,92,93]. In addition, SA can enhance plant resistance to oxidative stress. For example, research has also shown that SA can activate the expression of antioxidative enzymes, thereby enhancing the plant’s ability to eliminate reactive oxygen species and reduce cellular oxidative damage [94].

SA is an essential signaling molecule that triggers responses to a range of abiotic stressors, including cold, heat, drought, salt, heavy metals, and so on, and it also participates in developmental signaling pathways essential for normal plant growth, such as flowering and senescence [95,96].

Recent studies have found that SA can enhance plant tolerance to salt stress [43,47,82,84,97]. SA can regulate ion balance [83], maintain cell membrane integrity and stability, and modulate the antioxidative system to enhance the plant’s response to salt stress. In addition, other relevant studies have also demonstrated the important regulatory role of SA in plant drought tolerance [11,32,33,71,87,98]. Therefore, SA can enhance plant drought tolerance by regulating root growth, improving leaf water use efficiency, stabilizing cell membranes, and modulating the antioxidative enzyme system. Meanwhile, relevant studies also indicated that the biosynthesis of SA also exerted certain regulatory effects on normal plant growth and development, which is beneficial to the enhancement of plant tolerance to various kinds of stress [99,100,101,102].

As for SA’s roles in plant immunity, although many recent studies made certain advances in clarifying physiological mechanisms concerning SA’s involvement into plant tolerance responses to pathogens [103,104,105,106,107,108], in-depth studies regarding the further deciphering of relevant molecular mechanisms still need to be conducted.

As further research into the molecular mechanisms of SA continues, we will continue to gain deeper understandings of plant stress tolerances. Therefore, this paper highlights the significance of SA in enhancing plant resilience against both abiotic and biotic stresses, thereby promoting their survival and productivity in challenging environmental conditions (Figure 1). Further studies in this field can contribute to the development of effective strategies for crop improvement and stress management [109].

During their growth and development, plants also encounter various combinations of stresses (including multiple abiotic or biotic stresses, or both abiotic and biotic stresses) simultaneously, which frequently exert more severe damaging effects on plants than any single stress. Numerous recent studies suggested that SA can exert certain regulatory or alleviative effects on multiple stresses simultaneously, which are beneficial for plant tolerance responses to these stresses [110,111,112,113,114,115]. On the other hand, it was shown that without stress, SA can maintain normal plant growth and development [116,117], whereas under biotic stress, SA-mediated signaling was elicited to activate plant innate immunity responses [117], thereby suggesting a possible link between SA’s regulatory roles in both plant growth and immunity.

In a word, recent studies made remarkable advances in SA’s roles in plant tolerance responses to both biotic and abiotic stresses [47,48,71,72,82,84,86,87,88,118,119,120,121,122,123], and in both physiological [124,125,126] and molecular perspectives [127,128,129,130]. For example, Yao et al. proposed a high air humidity-triggered significant suppression of SA accumulation and signaling associated with compromised immunity of *Arabidopsis* plants [130], which indicates the value of research concerning SA’s roles in multiple stresses confronted by plants. Therefore, this paper aims to provide theoretical references for further studies concerning the clarification of molecular mechanisms for SA’s roles and the improvement of crop yield and management under stress.

## 2. Biosynthetic Pathway of SA in Plants

Two distinct pathways are involved in the biosynthesis of SA in plants: the isochorismate synthase (ICS) pathway and the phenylalanine ammonia-lyase (PAL) pathway [25,78,99,100,101,102]. The ICS pathway was primarily elucidated using *Arabidopsis* mutants that were deficient in pathogen-induced SA biosynthesis, while the PAL pathway has been found to contribute to SA biosynthesis in tobacco [78].

Research has proposed and reviewed that the levels of SA in plants are regulated by both positive and negative feedback. Under positive feedback regulation, genes involved in SA biosynthesis (ICS1/EDS5/PBS3) and N-hydroxypipecolic acid (NHP) biosynthesis (ALD1/SARD4/FMO1) are coordinately regulated by the immune transcription factors SARD1 and CBP60g. Activation of these factors leads to increased SA and NHP biosynthesis. Upstream of SARD1 and CBP60g, TGA1/TGA4 and GTL1 act as positive regulators, while WRKY70 and CAMTA1/2/3 act as negative regulators of gene transcription related to SA and N-hydroxypipecolic acid (NHP) biosynthesis [78,79]. Additionally, calcium ions (Ca^2+^) may play a dual role in regulating SA and NHP biosynthesis. They positively modulate the activities of calcium-dependent protein kinases (CPKs) and CBP60g, while negatively regulating the activities of CBP60a and CAMTA1/2/3. However, the exact mechanisms in this process are not yet fully understood [78].

## 3. The Role of SA in Plant Tolerance Responses to Biotic Stresses

Plants, including crops, are constantly exposed to various biotic stresses [103], such as pathogens (e.g., bacteria, fungi, oomycetes, nematodes), arthropods (particularly insects), birds, mammals, and competitive plants (especially weeds) [104].

In the recent decade, especially since 2021, numerous reviews and research articles concerning SA’s roles in plant tolerance responses to biotic stresses have been published, and some articles published from 2021 to 2023 and selected from four database sites, including ScienceDirect, Web of Science, Scopus, and MDPI, are listed in Table 1. From Table 1, it can be seen that these selected articles mainly deal with such biotic stress factors as bacteria, fungi, and insects.

It was reported that SA has the ability to activate various antioxidant enzymes, elicitor pathogenesis-related (PR) genes, and systemic acquired resistance (SAR) pathways, and even trigger programmed cell death (PCD) [3].

It was reviewed that SA acts as a vital endogenous signal for SAR, contributing to the activation of defense responses against biotrophic pathogens and enhancing plant resistance to multiple pathogens [4]. On the other hand, it was also considered that SA is involved in the activation of multiple pathways, including MAPK (mitogen-activated protein kinase), CDPK (calcium-dependent protein kinase), and other protein kinases, thus promoting the biosynthesis of secondary metabolites in plants, which are crucial in plant–pathogen interactions [6].

Vañó et al. proposed that SA, as a crucial phytohormone with a special role in plant defense responses to pathogens, is targeted by many clubroot pathogens, which actively attempt to suppress SA biosynthesis, accumulation, or downstream function [60]. Wang et al. reported that in apple, SA significantly upregulates NPR1 and PR1 and enhances endogenous SA levels. On the other hand, (E)-2-Hexenal is one of the most prevalent and abundant volatile substances in fruits. An (E)-2-Hexenal-based coating initiates the SA signaling pathway and induces an increase in resistance-related genes and enzymes [66].

In another study by Wang and Dong, it was discovered that the SA signaling pathway controls the synthesis of antimicrobial PR proteins, which are a critical mechanism against bacterial pathogens of plants [105]. Furthermore, Orozco-Mosqueda et al. proposed that bacterial pathogen infections elicit the secretion of SA-dependent PRs in plants, which directly contributes to the acquisition of SAR via the SA signaling pathway [106]. In table grapes, it has been shown that complex interplays among phytohormones and plant secondary metabolites, such as abscisic acid (ABA), indole-3-acetic acid (IAA), SA, and JA, play a crucial role in the fruit ripening and immune response or defense against necrotrophic fungal pathogens [107].

According to Tripathi et al., SA triggers systemic acquired resistance (SAR) in plants, leading to enhanced resistance against various pathogens [108]. Moreover, functional SA analogs have been found to induce plant defensive responses to pathogens by activating SA-mediated signaling pathways [108,109].

A combination of water deficit and nutrient deprivation can also cause certain damages comparable to those caused by pathogens. For example, the lack of visual pathogens was also reported in house leek (*Sempervivum tectorum* L.) individuals, and the content of SA was strongly positively correlated with leaf hydration and relative water content. On the other hand, it is possible that ABA and SA may interact antagonistically to modulate certain aspects related to morphological changes induced by water deficit [110].

As for SA’s roles in plant immunity, it was proposed that SA plays a critical role in plant immune signaling, contributing to the elicitation of the hypersensitive response (HR) and the activation of SAR [111]. Additionally, it was also reported that SA triggers the HR and SAR. SA synthesis rapidly follows pathogen detection through receptors such as PRRs and NBLRRs, leading to changes in cellular redox status and subsequent expression of immune-related genes [112].

Apart from its roles in plant immunity, SA can also simultaneously exert certain effects on plant tolerances to abiotic stresses by acting together with other molecules under biotic stresses. For instance, it was reported that biomolecular condensate (BMC) assembly can be triggered by the accumulation of stress-related metabolites, including SA, or changes in cellular redox state. Small molecules like SA can bind and modify the intramolecular interactions of proteins, thus participating in plant signaling responses to both biotic and abiotic stresses [113].

## 4. The Role of SA in Plant Tolerance Responses to Abiotic Stresses

In the recent decade, especially since 2021, numerous reviews and research articles relating to SA’s roles in plant tolerance responses to abiotic stresses have been published, and some articles published from 2021 to 2023 and selected from four database sites, including MDPI, ScienceDirect, Web of Science, and Scopus, are listed in Table 2. From Table 2, it can be concluded that these selected articles mainly focus on such abiotic stress factors as drought, salt, heat, cold, and heavy metals.

### 4.1. The Role of SA in Drought Stress

It is generally held that the following harms can be caused to plants and crops by drought stress: water deficiency, hindered transpiration, wilting and withering, nutrient imbalance, disrupted physiological metabolism, decreased yield, and increased biotic stresses.

During drought conditions, the supply of water in the soil decreases, making it difficult for plants to absorb enough water for normal growth and development, resulting in water deficiency [114].

As a consequence of biotic stresses, under drought conditions, plants become more susceptible to pest and disease infestation, as their defense mechanisms are weakened. The harms caused by drought stress significantly impact plant growth, development, crop yield, and quality [86,98].

As a crucial phytohormone, SA has been found to alleviate plant drought stress to some extent [85,87,98]. SA’s main functions in drought stress alleviation include: promoting seedling growth, enhancing plant antioxidant capacity, regulating plant water balance, promoting the expression of stress-related genes in plants, and regulating plant physiological metabolism [137]. For example, it was reported that priming wheat seeds with 50 µM SA was found to enhance drought stress tolerance by upregulating antioxidant defense and glyoxalase systems, thus ensuring better wheat seedling establishment [11]. SA at a concentration of 0.5 mM effectively mitigated the negative effects of drought stress on Sardari ecotypes of winter wheat. It was also found to improve photosynthetic performance, maintain membrane permeability, induce stress proteins, and enhance the activity of antioxidant enzymes [18].

In addition, under drought conditions, SA can improve important physiological indicators in wheat, including seed germination rate, plumule length, root length, and shoot length. The application of SA to wheat leaves can also enhance seedling vigor and indirectly improve water use efficiency by increasing levels of potassium (K^+^), calcium (Ca^2+^), and magnesium (Mg^2+^) in the roots. Additionally, SA can enhance chlorophyll content and membrane stability index (MSI), thereby boosting overall photosynthetic efficiency and improving yield and yield-related traits in drought-stressed wheat plants [34]. SA can also act as a positive regulator of drought tolerance by modulating antioxidant defense and reactive oxygen species (ROS)-scavenging responses [35].

On the other hand, SA can also exert certain effects on plant tolerance responses to drought in combination with other stresses. For example, it was reported that under combined drought and heat stresses, SA levels were correlated with increased amino acid levels in certain citrus plants [36]. It was also reported that during the flowering stage, a physiological concentration of SA (1 mM) is significantly effective against salt stress. During the mature vegetative stage, proline accumulates under SA treatment, which is useful in developing NaCl-induced drought stress tolerance [44].

SA participates in the regulation of plant physiological metabolic processes, such as regulating stomatal opening and closure [137], thereby alleviating the adverse effects of drought stress on plants. For example, a study indicated that in rice, SA regulated stomatal aperture through the OsWRKY45-reactive oxygen pathway under salt and drought stresses [115] (Table 2).

### 4.2. The Role of SA in Salt Stress

Among the factors limiting crop production, soil salinization is a major environmental challenge. It is generally held that some phytohormones can exert certain promoting effects on plant tolerance to adverse environmental conditions, including salinity. For example, according to Fu et al. phytohormones such as ABA, auxin, cytokinin (CK), brassinosteroids (BR), JA, gibberellins (GA), SA, and ETH play certain roles in enhancing crop salt tolerance [43].

The harm of salt stress to plants includes the following aspects: water imbalance, disruption of ion homeostasis, impaired nutrient absorption, damages to photosynthesis, and oxidative stresses, thus resulting in inhibited plant growth and development [42,43,47,82,84,97,136,137,138,139].

SA is an endogenous substance produced by plants, and it plays a role in alleviating salt stress in plants [136,137,138,139]. There are several main mechanisms by which SA alleviates plant salt stress: ion balance regulation, antioxidant effects, and hormone level regulation [136,137]. For example, Hundare et al. reported that treatment with SA (a foliar spray of 0.5 mM SA) enhances the contents of chlorophyll a, chlorophyll b, total chlorophyll, and carotenoids, while reducing chlorophyllase activity under salt stress conditions. Additionally, SA treatment further increases the activities of antioxidative enzymes, including SOD, CAT, and POD, which are induced by NaCl stress [24]. According to Shaki et al., exogenous SA improves the response of safflower to salinity by increasing glycine betaine, total soluble protein, carbohydrates, chlorophylls, carotenoids, flavonoid, and anthocyanin contents. SA application under saline conditions decreases the levels of proline, indicating successful acclimatization of these plants to saline conditions [45]. According to Jini and Joseph, SA application significantly increases the decreased rates of germination and growth (in terms of shoot and root lengths) caused by salt stress. The treatment of SA to the high and low saline soils enhances the growth, yield, and nutrient values of rice. SA application also reduces the accumulation of Na^+^ and Cl^−^ ions caused by salt stress and at the same time increases the activities of antioxidant enzymes [46]. On the other hand, according to Ali et al., both individual and combined applications of plant growth-promoting rhizobacteria (PGPR) and SA alleviated the negative effects of salinity and improve all the measured plant attributes. The response of PGPR + SA was significant in enhancing the shoot and root dry weights, relative water contents, chlorophyll a and b contents, and grain yield of maize under higher salinity level [47].

In terms of specific effects, SA can promote the growth and development of plant roots, enhance plant stress resistance, increase chlorophyll content and photosynthetic efficiency in leaves, and reduce transpiration and water loss [48,83,98]. Additionally, SA can promote plant physiological metabolism [36,94,96,102,111,117], thus increasing plant disease resistance and improving salt and drought tolerance. Overall, SA helps plants adapt to salt stress environments and mitigate the damage caused by salt stress [136,137,138,139]. Other mechanisms by which SA enhances plant salt tolerance are mainly as follows: activation of stress response pathways [96,118,119,120,121,122], induction of heat shock protein synthesis [40], and regulation of the expression of stress response genes [103,123].

As an endogenous substance in plants, SA can enhance the plant’s tolerance by activating drought and salt stress response pathways when subjected to external stresses such as drought and salt stress [140,141]. SA can activate protein kinases and kinase cascades, promote the expression of stress-resistant genes, and thus increase the plant’s tolerance [15,116].

In summary, SA enhances plant salt tolerance through molecular biological mechanisms such as activating drought and salt stress response pathways, promoting the synthesis of protective enzymes, regulating ion channel activity, inducing modulating the expression of stress response genes [136].

### 4.3. The Role of SA in Heavy Metal Stress

A wide range of heavy metal ions, such as Pb^2+^, Cd^2+^, Hg^2+^, Cr^2+^, Cu^2+^ and so on, can cause severe damages to plants due to their toxic effects. Heavy metal pollutions exert toxic effects on plants mainly through the following aspects: inhibition of plant normal growth and development, suppression of plant absorption and utilization of nutrition, disruption of cellular structures and functions, accumulation of heavy metal ions in plants, and other toxic effects [49,50,51].

Heavy metal ions can exert potent inhibitory effects on plant growth and development by disrupting physiological processes concerning metabolism, and suppressing photosynthesis, respiration and transpiration. Heavy metal ions can disrupt plant absorption and utilization of nutrition. Some heavy metal ions can directly compete with the nutritious elements essentially needed by plants, thereby making it hard for plants to absorb and utilize nutrition and limiting their normal growth.

Numerous studies in plant physiology and molecular biology have indicated that both exogenous and endogenous SA exert alleviative effects on plant responses to various abiotic stresses, with a particular focus on heavy metal stresses. This effect is achieved through direct or indirect enhancement of their tolerances [31,49,50,51,52]. For example, according to Ur Rahman et al., the exogenous application of SA to plants induces acclimatization impacts by improving resistance and leading to tolerance against heavy metal stress. This is achieved through the activation of stress signaling hormonal pathways and the overexpression of stress-related genes and enzymes [49]. According to Saini et al., the exogenous application of SA has been reported to restore heavy metal-induced damage to various photosynthetic traits in plants, thereby improving their stress tolerance. Another major mechanism involved in SA-regulated heavy metal stress tolerance in plants is the modulation of ROS detoxification machinery to protect the plants against oxidative stress. SA has also been documented to modulate the levels of non-enzymatic antioxidant metabolites, including proline, upon exposure to heavy metal stress [51].

It was widely reported and reviewed that SA exerts certain alleviative effects on the toxicity of heavy metals. SA can enhance the activities of antioxidant enzymes (including SOD, POD and CAT), thus lowering the contents of intracellular and intercellular reactive oxygen species (ROS) and reducing oxidative damages caused by heavy metal ions in plants. According to Nivedha et al., SA can exert certain mitigating effects on cadmium toxicity [57]. According to Jia et al., exogenous application of SA has been found to significantly reduce cadmium (Cd) accumulation in tomato plants and alter its distribution. SA pretreatment enhances cell wall polysaccharide synthesis and related gene expression, leading to cell wall thickening and blocking Cd penetration. Additionally, SA decreases pectin methylesterase activity and content, reducing cell wall Cd accumulation and altering the Cd partition ratio [90].

Although SA at certain concentrations exerts some alleviative effects on stresses by heavy metal ions, SA at high concentrations also brings about some toxic effects to plants. Therefore, appropriate SA applications into the enhancement of plant tolerances to stresses should be taken into considerations in terms of both SA concentrations and patterns for applications.

### 4.4. The Role of SA in High-Temperature Stress

High temperatures or heat can have significant harmful effects on plant growth and development [2,21]. The main harms of high-temperature stress on plants include the following several aspects: impairment and reduction of photosynthesis, increased respiration, water stress, cell membrane damage, DNA damage, protein denaturation, reduced nutrient uptake, delayed flowering and fruit set, increased susceptibility to diseases and pests, and other physiological and biochemical changes.

High-temperature stress can also lead to a series of physiological and biochemical changes within the plant, such as imbalances in the antioxidant system, abnormal levels of endogenous hormones, and accumulation of reactive oxygen species, affecting the plant’s adaptability to the environment. Therefore, it is crucial for farmers and gardeners to monitor and mitigate these effects to ensure optimal plant health and productivity.

SA is an important endogenous signaling molecule in plants and crops, which plays a role in alleviating heat stress [33,74,82,86,87]. Its molecular biological mechanisms mainly include the following aspects: regulation of reactive oxygen species (ROS) homeostasis, accumulation of antioxidants [58,85], SA’s involvement in epigenetic regulation, and SA’s participation in hormonal signaling [74,87].

SA can maintain the cellular ROS homeostasis by regulating the accumulation and clearance of reactive oxygen species. Under heat stress conditions, the production of ROS increases significantly, leading to oxidative damage in cells. However, SA can effectively remove ROS and alleviate cell damage. Heat stress induces the synthesis and accumulation of antioxidants in plants, enhancing the cell’s antioxidant capacity. SA can promote the synthesis and accumulation of antioxidants by regulating the expression of related genes, thereby enhancing the plant’s tolerance to heat stress [11,82,86,87].

It is generally considered that SA plays a regulatory role in mitigating the damage caused by high-temperature stress. It acts through various molecular mechanisms, including enhancing the antioxidant system, regulating the synthesis of resistance proteins, modulating signal transduction pathways, and regulating ion balance. SA alleviates high- temperature stress in plants through mechanisms that enhance antioxidant capacity, increase resistance protein synthesis, regulate signal transduction pathways, and maintain ion balance.

These mechanisms collectively enable plants to better cope with the damage caused by high temperatures. According to Balfagón et al., under combined drought and heat stresses, SA levels were correlated with increased amino acid levels in certain citrus plants [36]. According to Rasheed et al., both NO and SA protect plants against heat stress by enhancing sulfur assimilation, reducing oxidative stress, and increasing the activity of antioxidant enzymes. Supplementation of NO or SA along with sulfur under heat stress recovers losses, improves photosynthesis and growth [42]. According to Nivedha et al., SA has been reported to reduce heat stress-induced ROS production and its subsequent oxidative damage [57]. According to Li et al., under heat stress, transcripts of several SA biosynthesis and signaling genes, as well as heat stress-responsive genes, were significantly elevated in SO_2_-pretreated maize seedlings. This increase in endogenous SA levels activated the antioxidant machinery and strengthened the stress defense system, improving the thermotolerance of the seedlings [73].

It should be noted that as an endogenous signaling molecule, the mechanism of action of SA is very complex and is regulated by multiple environmental and genetic factors. Different plants and crops may also exhibit variations in their response to SA. Therefore, further research is needed to gain a deeper understanding of the molecular biological mechanisms of SA in alleviating heat or high-temperature stress.

### 4.5. The Role of SA in Low-Temperature Stress

Low-temperature extremes (cold stress, including chilling 0–15 °C and freezing < 0 °C temperatures) limit plant growth and development and severely affect plant physiology and biochemical and molecular processes [38].

It is widely considered that SA plays a vital role in plant stress tolerance to a wide range of abiotic stresses [140,141], including low-temperature or cold stress [37,38,39,118]. In response to cold stress, SA can regulate plant metabolism and induce the biosynthesis of cold-resistant proteins, thus improving plant tolerance to low-temperature stresses. For instance, according to Seo et al., SA controls ROS levels by enhancing antioxidant enzymes during chilling stress, thereby increasing chilling tolerance. However, a high concentration of SA may lead to the accumulation of hydrogen peroxide, which reduces chilling tolerance [37]. According to Raza et al., SA has been found to regulate seed germination rate, early plant responses, and protein pattern in apoplasts, and enhance the antioxidative system, growth and photosynthetic attributes, net photosynthetic rate, and SOD, CAT, and POD activities under cold stress [38].

It is generally considered that SA is a crucial plant hormone that plays a significant regulatory role in the response of plants to low-temperature stress. Its mechanisms of action include the following: promotion of activation of the antioxidant system, preservation of cell membrane stability, regulation of gene expression, and promotion of low-temperature signal production.

In summary, the mechanisms by which SA alleviates plant cold and freezing stress involve the synthesis of cold-resistant proteins, regulation of signal transduction pathways, ROS scavenging, and modulation of hormone synthesis and signaling. These mechanisms interact and work together to enhance the plant’s adaptability to low-temperature stress.

### 4.6. The Role of SA in Other Abiotic and Biotic + Abiotic Stresses

Apart from SA’s roles in the abovementioned abiotic stresses, SA also plays a certain part in plant tolerance responses to other biotic stresses, such as arsenic stress [19], UV radiation-induced oxidative stress [57], waterlogging stress [94], and strontium stress [127] (Table 2 and Table 3). According to Khan et al., application of SA promotes photosynthesis and growth in plants, with or without arsenic stress, by improving plant defense systems and reducing oxidative stress through cross-talk with ethylene (ETH) and nitric oxide (NO) [19]. According to Nivedha et al., the mechanisms of radio-protective activity of SA-rich fractions were ROS inhibition and lipid peroxidation inhibition [57]. According to Hasanuzzaman et al., exogenous application of SA and kinetin (KN) reduces the content of malondialdehyde (MDA), hydrogen peroxide (H_2_O_2_), proline, and leaf electrolyte leakage under waterlogging stress. Additionally, under waterlogging stress, supplementation with SA and KN enhanced the activity of various antioxidant enzymes, such as ascorbate peroxidase (APX), monodehydroascorbate reductase (MDHAR), dehydroascorbate reductase (DHAR), glutathione reductase (GR), CAT, glutathione-s-transferase (GST), glutathione peroxidase (GPX), POD, glyoxalsse I (Gly I), and glyoxalsse II (Gly II), which are closely related to plant stress tolerance. Both exogenous SA and KN effectively improved ROS metabolism and enhanced waterlogging stress tolerance in soybeans by enhancing antioxidant defense and upregulating the glyoxalase system [94]. According to Pyo et al., under strontium stress, the expression of pathogenesis-related protein genes, including PR1, PR2, and PR5, were found to be related to SA in response to stress caused by strontium [127].

In the recent decade, especially since 2021, numerous reviews and research articles concerning SA’s roles in plant tolerance responses to both biotic and abiotic stresses have been published, and some articles published from 2021 to 2023 and selected from four database sites, including ScienceDirect, Web of Science, Scopus, and MDPI, are listed in Table 3. From Table 3, it can be seen that these articles mainly deal with plants stressed simultaneously by multiple biotic and abiotic stresses, which are of vital importance in crop improvement and stress management for greater yield.

## 5. Interaction of SA with Other Molecules in Plant Defense Responses

The interaction between SA and other small molecules or plant hormones can enhance or weaken the resistance of plants through various molecular biology mechanisms. Here are some possible mechanisms: SA’s regulation of the biosynthesis of other phytohormones, SA’s regulation of hormone signal transduction pathways, SA’s regulation of the expression of hormone-responsive genes, and competitions between SA and its interacting hormones.

SA may regulate the resistance of plants by modulating the synthesis of other plant hormones [12,110]. For example, SA has been found to inhibit the synthesis of ETH and the methylation of JA, thereby reducing the plant’s response to adversity [12,19,56,125,126,127].

SA can intervene in the signal transduction pathways of other plant hormones, thereby enhancing or weakening the plant’s tolerance to stress [56,70,127]. For example, SA can regulate the plant’s tolerance by activating or inhibiting hormone signal transduction pathways [125,128].

SA is a universally recognized signaling molecule that is considered a pivotal stress hormone in plants. It plays a crucial role in both normal plant growth and stress tolerance. Numerous studies have demonstrated that SA’s involvement in plant tolerance to abiotic and biotic stress is a vital mechanism for their normal growth, defense against unfavorable environmental factors and pathogens, and even survival. For instance, Yang et al. reviewed that SA has been found to confer abiotic stress tolerance in horticultural crops by acting as a signaling molecule that triggers various physiological and morphological responses [129].

SA and other plant hormones may compete to activate or inhibit shared components or signal transduction pathways, thereby regulating the plant’s tolerance to stress. This competition may result in dominant or complementary effects of hormone signals, affecting the plant’s response to adversity.

SA is also an important signaling molecule for triggering responses to a myriad of abiotic stresses, including cold, heat, drought, ultraviolet, heavy metals, and so on, and participates in developmental signaling pathways that control and regulate many physiological and biochemical processes in normal plant growth, development, flowering, ripening, and aging [8,27,28,29,53,54,58,59,72,91,96].

For example, according to Das et al., Zinc-chitosan-SA (ZCS) nanoparticles enhanced morpho-physiological features, photosynthetic pigment status, osmotic status, osmoprotectant synthesis, ROS-scavenging enzyme activity, membrane integrity, cellular protection, and yield increment during drought stress [33]. According to Malko et al., the use of multiple chemicals, including SA, Zn, polyamines (PA), sodium nitroprusside (SNP), etc., with pre- and post-emergence treatment, has enhanced drought tolerance and maintained the normal physiology of wheat under changing climates [34]. According to Lafuente and Romero, citrus fruits’ heat-induced cross-adaptation to chilling is regulated by phytohormones such as JA, JA-related metabolites, SA, IAA, and ABA [39]. According to Wei et al., SA and auxin exerted antagonistic regulatory effects on the transcripts of the MeHsf8-MeHSP90.9 module, which is closely related to plant immunity. MeHSP90.9 interacted with MeSRS1 and MeWRKY20 to activate SA biosynthesis while inhibiting auxin biosynthesis, thus providing a mechanistic understanding of MeHSP90.9 co-chaperones in plant immunity [40]. According to Afzal et al., SA and moringa leaf extract (MLE) treatments recorded the maximum number of productive tillers and biological yield. SA treatment maximally enhanced free proline levels and protein content in wheat cultivars [41]. According to Rasheed et al., both NO and SA protect plants against heat stress by enhancing sulfur assimilation, reducing oxidative stress, and increasing the activity of antioxidant enzymes. Supplementation of NO or SA along with sulfur under heat stress recovers losses and improves photosynthesis and growth [42]. According to Mutlu-Durak et al., although SA can enhance salinity tolerance, the beneficial effects observed with willow extract applications cannot be explained by just their SA content. The same benefits could not be obtained with the exogenous application of pure SA [48]. According to Khalil et al., SA, alone and in combination with kinetin or calcium, has been found to improve growth traits, photosynthetic pigments, carbohydrate contents, and nitrogenous constituents in *Phaseolus vulgaris* plants. Additionally, plants cultivated from seeds soaked in SA in combination with kinetin or calcium showed enhanced activities of antioxidant enzymes and proline accumulation under nickel and/or lead stress [50].

According to Thepbandit et al., the use of SA-Ricemate as a foliar spray at certain concentrations can reduce the severity of bacterial leaf blight disease by 71%. In field conditions, SA-Ricemate significantly reduced disease severity by 78% and increased total grain yield [67]. According to Kotapati et al., supplementation with both SA and SNP significantly reduced the toxic effect of Ni and increased root and shoot length, chlorophyll content, dry mass, and mineral concentration in Ni-treated plants. Exogenous application of SA or SNP, specifically the combination of SA and SNP, protects finger millet plants from oxidative stress observed under Ni treatment [76]. According to Talaat and Hanafy, co-application of SA and spermine (SPM) can be a superior method for reducing salt toxicity in sustainable agricultural systems. Exogenously applied SA and/or SPM relieved the adverse effects caused by salt stress and significantly improved wheat growth and production by inducing higher photosynthetic pigment content, nutrient acquisition, ionic homeostasis, osmolyte accumulation, and protein content [82,84].

According to Munsif et al., co-application of SA and K^+^ under drought stress conditions significantly mitigated negative effects induced by drought stress. This combined treatment has been shown to improve grain yield and enhance water use efficiency, particularly under mild and severe drought stress, respectively. Co-utilization of SA and K^+^ helps regulate osmotic and metabolic processes, stabilize cell components, and ultimately overcome the detrimental impacts of drought stress [85]. According to Alotaibi et al., the co-application of essential plant nutrients and SA was proven to be a feasible, profitable, and user-friendly strategy for mitigating the negative effects of deficit irrigation stress. This approach has also demonstrated further improvements in the growth and production of wheat under normal irrigation conditions [86]. According to Zhang et al., MYB transcription factors played important roles in various physiological activities of plants, including phytohormone signal transduction and disease resistance. Transcripts of MiMYB5, −35, −36, and −54 have shown positive responses to early treatments of SA, methyl jasmonate (MeJA), and hydrogen peroxide (H_2_O_2_), suggesting their involvement in these signaling pathways [88]. According to Li et al., in plant immunity, the hormones SA and JA played mutually antagonistic roles in resistance against biotrophic and necrotrophic pathogens, respectively. Transgenic plants with a synthetic dual SA- and JA-responsive promoter controlling the expression of an antimicrobial peptide exhibited enhanced resistance to a wide range of pathogens [93].

According to Kumar and Giridhar, exogenous applications of SA and MeJA were shown to alleviate the maturation-triggered repression of N-methyltransferase (NMT) genes, which play certain regulatory roles in plant growth, development, and stress responses and are of vital importance for plant survival and adaptability. This suggests a potential cross-talk between signaling cascades involved in the regulation of NMT gene expression during maturation and the overexpression induced by SA and MeJA treatment in the mature endosperm [128].

In summary, the interaction between SA and other plant hormones enhances or weakens the plant’s tolerance through mechanisms such as the regulation of hormone synthesis, signal transduction pathways, gene expression, and hormone competition. The specific details of these mechanisms will depend on the plant species, type of stress, and the context of hormone interactions.

## 6. Conclusions and Perspectives

Various mechanisms are widely known to alleviate plant stress tolerance at the molecular level. For instance, SA can induce defense genes in plants by activating the expression of a wide range of defense-related genes, including pathogenesis-related (PR) genes, which encode proteins involved in plant defense responses. PR proteins strengthen the plant’s immune system and confer resistance to pathogens or pests.

Meanwhile, SA can also regulate other phytohormones. For example, SA interacts with other phytohormones, such as JA and ETH, to modulate plant stress responses. It often acts synergistically or antagonistically with these hormones, depending on the stress type. This cross-talk helps plants fine-tune their defense strategies to combat specific stresses effectively.

In addition, SA primes plant immune systems by pre-activating defense mechanisms before stress onset. This priming induces a faster and stronger defense response when a stressor is encountered, providing a higher level of stress tolerance. It is also generally held that SA can activate systemic acquired resistance (SAR) because SA is a key player in SAR, a long-lasting defense response that occurs throughout the entire plant system following a localized pathogen attack. SAR confers enhanced resistance to a broad range of pathogens by activating defense responses in uninfected parts of the plant.

On the other hand, stomatal behavior, especially stomatal closure, is generally considered as an important physiological index for measuring plant tolerance [137]. It is generally held that SA can modulate stomatal behavior and influence stomatal closure [137], thus reducing water loss from the plant under stress conditions. This modulation helps conserve water and maintain plant hydration during drought or other stressors [92,115].

Meanwhile, a wealth of research at the molecular level on SA signaling pathways has indicated that in plants, SA-SABP complexes formed by SA conjugating with SABPs (SA- binding proteins) spread cellular messages to secondary messengers such as H_2_O_2_ and Ca^2+^. These messages are amplified by self-feedback mechanisms and transduced, triggering cellular responses activating a cascade of transcription factors, including NPR1, TGA, and WRKY. This, in turn, promotes the interaction among these factors and the expression of PR genes that confer SAR to plants.

For example, Zhu et al. demonstrated that localized infection by an avirulent pathogen leads to an increase in SA levels, which then activates NPR1 (nonexpressor of pathogenesis-related genes 1) to induce SAR and promote cell survival to prevent the spread of cell death signals. In response to varying levels of SA, two other SA receptors, NPR3 and NPR4, act together to maintain optimal NPR1 levels by degrading NPR1, thereby constraining cell death signals to localized levels. Additionally, methyl-SA (MeSA) functions as a mobile signaling molecule for SAR, and activation of SAR confers resistance to the entire plant against secondary infection by a virulent pathogen in systemic leaves [104].

Contrarily, Ding et al. proposed that as SA receptors participate in SA-mediated signaling pathways for plant immunity, NPR1 plays an opposing role to that performed by NPR3 and NPR4 [79].

Overall, SA’s alleviative effects on plant stress tolerance at the molecular level involve gene regulation, modulation of phytohormonal responses, strengthening of antioxidant defense capacity, immune system priming, activation of systemic acquired resistance, and regulation of stomatal behavior. These mechanisms collectively contribute to enhanced stress tolerance and improved plant health.

Numerous studies have also found that SA, as an endogenous signaling molecule in plants, can regulate plant stress tolerance under environmental stress. SA enhances plant resilience by modulating the activity of a series of antioxidant enzymes, regulating plant growth and development, and controlling gene expression. Research showed that SA can activate the plant’s immune response by regulating the expression of a series of disease-resistant genes, thereby enhancing the plant’s ability to resist diseases. For example, Luo et al. reported that FaSnRK1α regulates the expression of FaPAL1 and FaPAL2, affecting the content of SA. This regulation, along with the interaction with the FaWRKY33.2 transcription factor, increased the resistance of strawberry fruit to *Botrytis cinerea* through the SA signaling pathway [119].

Hui et al. reported that bacterial pathogens Xoo and Xoc were found to activate the expression of miR156 and miR529, which subsequently target and cleave the mRNAs of three self-interacting proteins, namely, OsSPL7, OsSPL14, and OsSPL17. The downregulation of these three OsSPLs leads to insufficient activation of their downstream target genes, OsAOS2 and OsNPR1. As a result, the levels of JA decrease, compromising the SA signaling pathway. This compromised SA signaling facilitates the proliferation of Xoo or Xoc, ultimately causing susceptibility in rice plants [121].

It is generally held that SA also plays an important regulatory role in plant growth and development. The latest research have indicated that SA can regulate seed germination, seedling growth, and plant establishment.

It was reviewed and demonstrated that SA is involved in stress-induced PCD in plants. Research indicates that SA can induce apoptosis in plant cells, thereby enhancing plant defense against adverse conditions.

Overall, research on the involvement of SA in plant stress resistance demonstrates its importance in regulating plant stress resistance, immune system, growth and development, and stress-induced programmed cell death. These research findings contribute to the understanding of the physiological regulation mechanisms of SA in plants and provide a theoretical basis for further improving plant stress tolerance.

As an important plant signaling molecule, SA plays a crucial regulatory role in plant stress tolerance. Through regulation of multiple cell signaling pathways, it is involved in plant responses to diseases, drought, and salt stress. In-depth research on the cell signaling pathways involving SA will help us better understand the mechanisms of plant stress responses, providing a theoretical basis for enhancing plant adaptability and genetic improvement of crops.

Genomics and proteomics are important tools for understanding the role of SA in alleviating plant stress. Genomics techniques, such as RNA-seq and microarray technology, can identify stress-regulated genes and examine how SA affects their expression. On the other hand, proteomic studies utilizing protein mass spectrometry can analyze the changes in the plant proteome under stress conditions regulated by SA. By comparing the protein expression profiles of SA-treated and control groups, proteins regulated by SA can be identified. These findings can further reveal the mechanisms of plant stress response and the involvement of SA.

In addition, bioinformatics analysis is crucial in processing the data generated by genomics and proteomics. Integrating these datasets allows for a deeper understanding of how SA alleviates plant stress, including the activation of specific pathways and involvement in signaling cascades. For instance, it enables the identification of SA-related regulatory factors and the prediction of stress-responsive genes regulated by SA.

It is worth noting that the role of SA in plant tolerance responses can vary depending on the specific stressor, plant species, and other factors. Additionally, the exact mechanisms involved in SA-mediated stress responses are still being investigated, and further research is needed to fully understand its complexities.

In the future, genomics and proteomics will continue to be indispensable in studying the mitigation of plant stress by SA because they can pinpoint the genes and proteins associated with SA and elucidate the mechanisms through which it regulates stress response.

## Figures and Tables

**Figure 1 plants-12-03475-f001:**
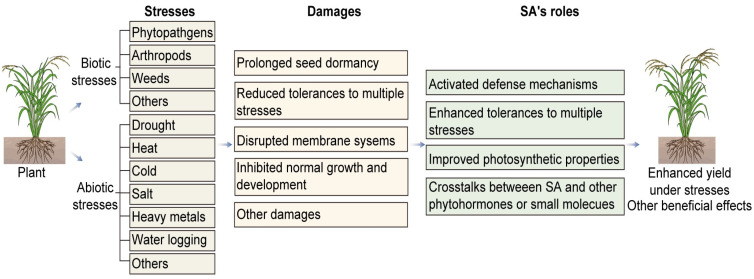
A diagram of SA’s roles in plant stress responses.

**Table 1 plants-12-03475-t001:** SA’s roles in plant biotic stress responses.

Stress Factors	Plant Species or Categorizations	SA’s Roles or Other Signaling Molecules’ Effects on SA	References
mycotoxin Fumonisin B1	*Arabidopsis* and various crops	Activation of various antioxidant enzymes, PR genes, SAR pathways, and PCD	[3]
multiple pathogens	general plants	Activation of SAR pathways; defense response against biotrophic pathogens	[4]
multiple biotic stresses	general plants	Activation of MAPK, CDPK, and other PKs; enhancement of SM biosynthesis	[6]
*Pectobacterium carotovorum*	Chinese cabbage	Possibly no participation in the defense response to *Pectobacterium carotovorum* infection	[12]
pathogens	*Arabidopsis*	Modulation of SA biosynthesis by SABC1’s role as a molecular switch in balancing plant defense and growth	[25]
pathogens	cassava	Activation of SA biosynthesis by MeHSP90.9’s interaction with MeSRS1 and MeWRKY20	[40]
bacterial pathogens	jujube	Indication of antimicrobial activity	[55]
pathogens	cruciferous crops	Disruption of SA biosynthesis and accumulation by biotrophic pathogens to hijack plant cellular processes	[60]
insect herbivores and pathogens	general plants	Defense responses of SA induction to pathogens or aphid feeding	[61]
pathogens	*Arabidopsis thaliana*	Role of *T. hamatum* as an inter-plant communicator as a consequence of antagonism between JA and SA	[62]
fungal pathogens	apple	Significant upregulation of NPR1 and PR1; enhancement of endogenous SA; initiation of SA signaling pathway by (E)-2-Hexenal-based coating	[66]
bacterial pathogens	rice	Reduction of disease severity; significant increment of total grain yield	[67]
weeds	crops	Alteration of crop growth by weeds through increment of SA signaling process	[68]
pathogens	wheat	Drastic change of contents of compounds involved in biosynthesis and metabolism of SA in roots	[74]
pathogens	general plants	Possible regulation of SA synthesis by cAMP level; stomatal closure by SA, NO, and cGMP	[92]
pathogens	tobacco and *Arabidopsis*	Antagonism of SA and JA in resistance to biotrophic and necrotrophic pathogens	[93]
pathogens	*Arabidopsis*, poplar	Pathogen-stimulated SA synthesis; SA-mediated defense; SA’s interplay with other hormones	[99]
pathogens	general plants	Basal defense and amplification of local immune responses; establishment of SAR; regulation of plant immunity	[102]
bacterial pathogens	kiwifruit vines	Induction of SA-pathway defense genes by postharvest ASM	[103]
pathogens	general plants	Establishment of plant immunity; elicitation of defense responses; establishment of both local and systemic resistance against various pathogens	[104]
pathogens	general plants	Elicitation of SAR through signals involving SA	[106]
pathogens	table grape	Correlation of SA accumulation and degree of anthracnose disease resistance and symptoms; resistance induction by SA level increase	[107]
pathogens	general plants	Elicitation of HR; activation of SAR; responses of SA synthesis to pathogens via receptors	[112]
pathogens	strawberry	Effects of *FaSnRK1α* on SA content through regulation of *FaPAL1* and *FaPAL2* expressions; increment of pathogen resistance	[119]
pathogens	rice	Rice susceptibility caused by proliferation of bacterial pathogens facilitated by compromised SA signaling	[121]
pathogens	grapevines	GA’s role as a component of the grapevine SA-dependent response	[123]
pests	grapevines	Reduced injury by JA and SA application; reduction of number of eggs laid by *D. suzukii* females in JA- and SA-treated plants	[125]

Abbreviations: SA (salicylic acid), PR genes (pathogen-related genes), SAR (systemic acquired resistance), PCD (programmed cell death), SM (secondary metabolites), PK (protein kinase), MAPK (mitogen-activated protein kinase), CDPK (calcium-dependent protein kinase), SABC1 (salicylic acid biogenesis controller 1), JA (jasmonic acid), PR1 (pathogenesis-related genes 1), NPR1 (nonexpressor of pathogenesis-related genes 1), cAMP (cyclic adenosine monophosphate), NO (nitric oxide), cGMP (cyclic guanosine monophosphate), ASM (acibenzolar-S-methyl), HR (hypersensitive response), GA (gibberellins).

**Table 2 plants-12-03475-t002:** SA’s roles in plant abiotic stress responses.

Stress Factors	Plant Species or Categorizations	SA’s Roles or Other Signaling Molecules’ Effects on SA	References
drought	wheat	Elicitation of drought stress tolerance; better seedling establishment	[11]
wounding	*Arabidopsis* and maize	Mediation of glutamate receptors in regeneration; anti-correlation between increased SA response and regeneration in older tissue	[14]
drought	winter wheat	Effective alleviation of drought stress	[18]
arsenic stress	rice	Promotion of photosynthesis and growth through cross-talk with ETH and NO	[19]
fomesafen toxicity	sugar beet	Alleviation of fomesafen stress, improvement of some photosynthetic indexes in leaves, maintenance of cell membrane integrity, and amelioration of adverse effects of fomesafen on seedling growth	[20]
high temperature	maize	Reduction of decrease in dry matter and leaf area index; alleviation of chloroplast ultrastructure disruption, and decrease in photosynthetic rate and chlorophyll content; promotion of antioxidant enzyme activity and counteraction of increase in JA and ABA contents	[21]
salt	ginger	Enhancement of chlorophyll and carotenoid contents; further increase in NaCl-induced antioxidative enzymatic activities	[24]
drought	canola	Development of stress tolerance and improvement of plant growth by co-application of *Pseudomonas putida* and SA	[32]
drought	wheat	Amelioration of physiological and photosynthetic features; improvement of osmotic status; activation of ROS-scavenging enzymes; promotion of yield increment	[33]
drought	wheat	Refinement of drought tolerance; maintenance of normal physiology	[34]
drought	general plants	Positive regulation of drought tolerance phenotype	[35]
combined drought and heat stresses	citrus plants	Correlation of SA levels with the increase of amino acid levels	[36]
cold	pepper	Regulation of antioxidant metabolism to control ROS; enhancement of antioxidant enzymes to increase chilling tolerance	[37]
cold	*Arabidopsis* and crop plants	Regulation of seed germination rate, early plant responses, protein pattern in apoplasts; enhancement of antioxidative system, growth, and photosynthetic attributes	[38]
heat, cold	citrus plants	Involvement in fruits’ heat-induced cross-adaptation to chilling	[39]
heat	wheat	Protection against heat stress; reduction of oxidative stress; increment of antioxidant enzyme activities; improvement of photosynthesis and growth	[42]
salt	cereal crops	Enhancement of salt tolerance	[43]
salt, drought	chilli	Significant effects against salt stress; elicitation of proline accumulation for the development NaCl-induced drought stress tolerance	[44]
salt	maize	Alleviation of salinity; significant enhancement of grain yield	[47]
salt	maize	Enhancement of salinity tolerance	[48]
heavy metals	general plants	Induction of acclimatization impacts; improvement of tolerance against heavy metal stress	[49]
heavy metals	common bean	Improvement of growth traits and photosynthetic indexes; enhancement of antioxidant enzyme activities and proline accumulation	[50]
heavy metals	general plants	Improvement of stress tolerance; modulation of ROS detoxification machinery to protect plants against HM-induced oxidative stress; modulation of non-enzymatic antioxidant metabolite levels	[51]
drought	*Camellia* *oleifera*	Decline of SA level during drought stress; exhibition of stronger antioxidant capacity, water regulation ability, and drought stress protection by drought-treated group	[71]
heat	maize	Significant elevation of transcripts of SA biosynthesis and signaling, and heat stress-responsive genes in SO_2_-pretreated seedlings; improvement of seedlings’ thermotolerance	[73]
salt	sugarcane	Effective mitigation of adverse impact; facilitation of better germination and early seedling growth; maintenance of relative water content, membrane stability and overall plant growth	[77]
salt	wheat	Alleviation of plant growth inhibition; increment of N, P, and K^+^ acquisition; enhancement of endogenous SA and SPM levels by exogenous SA and/or SPM applications	[82]
salt	St John’s wort	Improvement of growth	[83]
salt	wheat	Salt stress alleviation by SA and/or SPM; significant improvement of growth and production	[84]
drought	wheat	Substantial reduction of drought influence, and enhancement of grain yield and water use efficiency by co-application of K^+^ and SA	[85]
dry climatic conditions	wheat	Attenuation of deficit irrigation, and improvement of growth and production by co-application of essential plant nutrients and SA	[86]
drought	wheat	Regulation of stress response	[87]
cadmium	tomato	Significant reduction of cell wall Cd accumulation; changes in Cd distribution	[90]
waterlogging stress	soybean	Reduction of some physiological indexes by SA and KN; enhancement of antioxidant defense by SA and KN; effective improvement of ROS metabolism and waterlogging stress tolerance	[94]
salt	cashew	Attenuation of salt stress; increment of photosynthetic pigment synthesis	[97]
drought	oregano	Improvement of PSII efficiency under moderate drought stress	[98]
water deficit and nutrient deprivation	*Sempervivum tectorum* L.	Strong positive correlation between SA and some physiological indexes; possible antagonistic modulation of ABA and SA on water deficit-induced morphological changes	[110]
water deficiency	potato	Mitigation of oxidative damage; alleviation of water deficiency stress	[114]
salt, drought	rice	Regulation of stomatal aperture through the OsWRKY45-reactive oxygen pathway; regulation of adaptation to soil salinity and drought stress	[115]
abiotic stresses	watermelon	Regulation of some ClBBX genes’ transcription for possible growth and development regulation	[118]
abiotic stresses	horticultural crops	Provision of abiotic stress tolerance; elicitation of various physiological and morphological responses to stress; regulation of stress-responsive genes’ expression; direct interaction with various hormones, proteins, and enzymes involved in abiotic stress tolerance	[129]
abiotic stresses	general plants	Provision of multiple abiotic stress tolerance	[131,132]
salt	general plants	Improvement of resistance to NaCl stress	[133]
deficient irrigation	patato	Significant enhancement of growth characteristics, yield components, and photosynthetic attributes	[134]
abiotic stress	horticultural crops	Improvement of productivity; reduction in oxidative injuries; elevation of photosynthetic attributes	[135]
saline stress	general plants	Induction of saline stress tolerance	[136]

Abbreviations: SA (salicylic acid), ETH (ethylene), NO (nitric oxide), JA (jasmonic acid), ABA (abscisic acid), ROS (reactive oxygen species), HM (heavy metals), SPM (spermine), Cd (cadmium), KN (kinetin), PSII (photosystem II), ClBBX (*Citrullus lanatus* B-BOX).

**Table 3 plants-12-03475-t003:** SA’s roles in plant biotic + abiotic stress responses.

Stress Factors	Plant Species or Categorizations	SA’s Roles or Other Signaling Molecules’ Effects on SA	References
multiple stresses	maize	Expression of SM biosynthesis pathway	[2]
multiple stresses	general plants	SA interplays with MT; cross-talks between MT, ETH, and SA in plant pathogen resistance; SA’s induction of plant defense responses against pathogens	[5]
multiple stresses	general plants	Freezing tolerance establishment; improvement of PSII stability; increased chlorophyll content and improved yield of rice plants with bacterial blight disease	[22]
multiple stresses	general plants	Partly independent role of SA and MeSA’s involvement in the development of SAR	[23]
environmental stresses	sorghum	SA’s participation in sorghum plant responses to abiotic and biotic stresses	[26]
abiotic and biotic stresses	mango	Positive responses of MiMYB transcription factor genes to early treatments of SA, MeJA, and H_2_O_2_	[88]
abiotic and biotic stresses	general plants	Evolution of MT, auxin, and SA from a common precursor, chorsimate	[109]
pathogens, heat	*Arabidopsis*	Restoration of SA production for plant immunity by optimized CBP60g expression at elevated growth temperature	[111]
abiotic and biotic stresses	general plants	Elicitation of BMC assembly by accumulation of SA; modification of proteins’ intramolecular interactions by SA binding	[113]
strontium stress, biotic stress	*Arabidopsis*	Relationship between expression of some PR genes and SA’s response to Sr stress	[127]
high humidity, biotic stress	*Arabidopsis*	Compromised plant immunity due to high air humidity-triggered suppression of SA accumulation and signaling	[130]

Abbreviations: SA (salicylic acid), SM (secondary metabolites), MT (melatonin), ETH (ethylene), PSII (photosystem II), MeSA (methyl salicylate), SAR (systemic acquired resistance), BMC (biomolecular condensate), Sr (strontium).

## Data Availability

Not applicable.

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
