# Peer review of "Advances in Roles of Salicylic Acid in Plant Tolerance Responses to Biotic and Abiotic Stresses"

_plants, 2023, doi:10.3390/plants12193475_

Round 1

Reviewer 1 Report

The manuscript reviews current research on the role of salicylic acid in plant resistance to abiotic and biotic stress factors. The manuscript is written with high quality and the authors follow the chosen plan.

This topic is relevant and undeniably important. However, every year many review articles are published on this topic, to which, however, the authors do not refer. For example, articles published in the current year (DOI 10.3389/fpls.2023.1141918; https://doi.org/10.3390/ijms24043388; https://doi.org/10.1016/B978-0-323-98332-7.00001-9) and earlier (2022: https://doi.org/10.1016/B978-0-323-91005-7.00008-4; 2021: Journal of Stress Physiology & Biochemistry, Vol. 17, No. 3, 2021, pp. 32- 50 ISSN 1997-0838; 2018: DOI: 10.5958/0974-360X.2018.00583.8, etc.).

The plan for presenting the material of the manuscript is very similar to that in the review article (Role of Salicylic Acid in Biotic and Abiotic Stress Tolerance in Plants, 2020, DOI: 10.1007/978-981-15-4890-1_23) and some other works.

Due to the large number of works published on this topic, it is very difficult to present original material.

The manuscript contains a good selection of literature, which the authors presented in tables. However, the reviewer felt that the data collected in Table 1 and described in the chapter on “biotic stress” could be structured and presented under subheadings.

The tables presented in the manuscript simply list the articles found by the authors. It is recommended to systematize and group these works, highlighting an important feature: plant species/variety, stress factor, or SA effect. At the moment, the text of the manuscript exists as if independently of the tables.

In addition, the tables are located far from the text describing them.

The reviewer recommends major revision of the manuscript.

Author Response

Dear reviewers,

Thank you very much for your kind comments and suggestions!We have revised the paper according to your comments and made the following revisions( please,see the attached).

Reviewer 2 Report

In the Review Article "Advances in Roles of Salicylic Acid in Plant Tolerance Responses to Biotic and Abiotic Stresses", the authors point out the various important roles of salicylic acid within various plants in supporting and strengthening their endogenous defense systems against many biotic and abiotic stresses. The review is interesting and clear with a valid and better selection of topics. The review also included well-presented data. However, revisions are needed as follows:

1. Line 14: “omnipotent” is an inappropriate word to describe the strength of SA, please change it to a suitable descriptive word.

2. Line 16: change “stresses” to “stress”.

3. Line 17: revise to “both biotic and abiotic stresses”.

4. Line 17: Please do not begin any sentence with an abbreviation (e.g., SA).

5. Line 22: change “stresses” to “stress”.

6. Line 22: “SA acts with other signaling molecules”, please briefly mention these other signaling molecules.

7. Line 23: the same comment in lines 16 and 22.

8. Line 23: change “tolerances” to “tolerance”, and revise the phrase to “improving tolerance under stress”.

9. Line 24: “tolerance” not “tolerances”.

10. Line 26: “stress” not “stresses”.

11. Line 27: Do not repeat title words in keywords.

12. Line 32: revise to “In order to adapt to stress, …..”.

13. Line 34: salicylic acid (SA), and only write SA in line 37.

14. Line 35: abbreviate salicylic acid to SA.

15. Line 41: Please add the following article to the literature:

Desoky, E.-S.M., Selem, E., Abo El-Maati, M.F., Hassn, A.A.S.A., H.E.E.
Belal, H.E.E., Rady, M.M., AL-Harbi, M.S., Ali, E.F. (2021). Foliar
supplementation of clove fruit extract and salicylic acid maintains the
performance and antioxidant defense system of Solanum tuberosum L. under
deficient irrigation regimes. Horticulturae 7: 435.

Abd El-Mageed, T.A., Semida, W.M., Mohamed, G.F., Rady, M.M. (2016).
Combined effect of foliar-applied salicylic acid and deficit irrigation
on physiological–anatomical responses, and yield of squash plants under
saline soil. South Afr. J. Bot. 106: 8–16.

Semida, W.M., Rady, M.M. (2014). Pre-soaking in 24-epibrassinolide or
salicylic acid improves seed germination, seedling growth, and
anti-oxidant capacity in Phaseolus vulgaris L. grown under NaCl stress.
J. Hortic. Sci. Biotechnol. 89 (3): 383–344.

16. Line 43: Just write the abbreviation (SA).

17. Line 46: revise the sentence.

18. Lines 47 and 48: resistances or tolerances should be resistance or tolerance.

19. Line 49: Just write the abbreviation (SA).

20. Line 53: Please do not begin any sentence with an abbreviation (e.g., SA).

21. In this way, please read the manuscript carefully and correct the errors as I indicated to you in the previous 20 points.

22. Figure 1: Part of the figure is missing from the page, please edit it.

23. Figure 1 is brief; it needs some detail.

24. Line 177: Adjust the beginning of the sentence.

25. The section on “Conclusions and Perspective” is very long.

25. In general, please check the manuscript for grammatical errors and typos.

26. The journal's guidelines must be followed regarding writing text and arranging graphs, tables, and references.

My recommendation: This Review Article deserves to be published in the "Plants", but the above points must first be taken into consideration.

Moderate English editing is required.

Author Response

Dear referees;

Thank you all your comments and kind suggestions! We have completely improved the paper according to your comments.All your marked places have been checked out and corrected. Please,see the attached response.

Reviewer 3 Report

The reviewed article addresses an interesting and important theme related to the involvement salicylic acid (SA) in mechanisms enabling adaptation of plants to stressful environments (both biotic and abiotic factors). It is a great collection of references to the articles describing importance of SA for protection of plants against pathogens, drought, salinity, heavy metals, extreme temperatures and other detrimental external factors. It will be useful for those, who deal with SA. However, I'm not sure it will attract a wide readership. To achieve wide readership, authors should work hard to revise and improve the text. Numerous repetitions of the same phrases should be avoided. Definition of what SA is and mentioning its involvement in antioxidant defense is repeated about 30 times each as if this is done for the first time. It looks as if authors, who wrote different sections, did not read what other authors wrote. Otherwise repetitions appear, since authors just rewrote what they found in different articles they cite without making any attempt to summarize information found in them. These repetitions should be deleted. Instead authors should decipher abbreviations and provide definitions of numerous processes just mentioned in the text without any explanations. Some phrases sound as if authors do not quite understand what they are writing about. There is often no logic in the narrative. In its present state the article is not worthy of publishing in such rated journal as Plants. My exact comments are as follows: 1.      Line 46-47. “active anti-pathogenic substances[1,60-65], AND anti-fungal substances[66]” – this phrase sounds as if fungi are not pathogens. Meanwhile term “pathogens” involves fungi. 2.      Line 47. “plants’ resistances or tolerances to biotic stresses caused by pathogens, fungi and pests.” – Terms resistance and tolerance are close enough and authors should decipher what each means. According to some definitions resistance is the host’s ability to limit pathogen multiplication and tolerance is the host’s ability to reduce the effect of infection on its fitness regardless of the level of pathogen multiplication. Something like this should be written in the article since “tolerance OR resistance” is repeated in the article several times. Authors should only keep in mind that resistance and tolerance have different meaning in the case of abiotic stresses. 3.      Lines 59-62. Word “role” is repeated 5 times in the sentence. It is better to write it once or twcie and then provide the list of stresses (role in drought (), salinity ()…). 4.      Figure 1. There should be closer relations between damages and the role of SA (how SA protects plants against each type of damages) 5.      Schemes illustrating isochorismate synthase (ICS) pathway and the phenylalanine ammonia-lyase (PAL) pathway should be provided. This will make the section look less short. 6.      Line 102. “SA biosynthesis (ICS1/EDS5/PBS3) and N-hydroxypipecolic acid (NHP) biosynthesis” – authors should either explain how SA is related to N-hydroxypipecolic acid (NHP) or delete the letter. 7.      “peer-reviewed review” – this phrase is repeated several times. Cannot it be just review? 8.      124-127. “It is generally accepted that SA-dependent signaling processes are involved in sys-temic acquired resistance (SAR) in plants, leading to the synthesis of pathogenesis-related (PR) proteins in response to pathogen infections. It was also reviewed that SA acts as a vital endogenous signal for SAR” – this one of numerous examples of repeating the same (involvement of SA in SAR) in the sentences following each other. The second sentence should be deleted. Instead I advise to explain in short what SAR means. Not merely to provide deciphering of abbreviation, but describe its mechanism in short. 9.      Lines 135-137. “Vañó et al. proposed that SA, as a crucial phytohormone with a special role in plant defense responses to phytopathogens, is targeted by many clubroot pathogens, which actively attempt to suppress SA biosynthesis, accumulation, or downstream function. During host colonization, various phytopathogens may also modulate phytohormones directly involved in host defense responses, including SA, jasmonic acid (JA), and ethylene (ETH) [60].” - Effect of pathogen on SA is described in both sentences. Repetition of similar information should be avoided. 10.  Line 138-139. “An E2H-based coating initiates the salicylic acid signaling pathway” – it should be explained what E2H means. E.g., it is E)-2-Hexenal, which is one of the most prevalent and abundant volatile substances in fruits. 11.  Line 143. PR – is deciphered for the second time. 12.  Line 145. “bacterial pathogen infections elicit the secretion of SA-dependent PRs” – it may be understood that PRs are secreted by pathogens. The sentence should be clarified. 13.  Line 154 – SAR is deciphered for the second time. 14. Lines 159 -160. “Therefore, it is possible that ABA and SA may interact antagonistically to modulate certain aspects related to morphological changes induced by water deficit [110].” – mechanism of this antagonistic interaction between ABA and SA should be explained. First, it is necessary to write something about ABA and its role in plants protection against stress factors. 15. Line 163. “elicitation of the hypersensitive response (HR) and the activation of systemic acquired resistance (SAR)” – it is important to mention in short what hypersensitive response entails, while it is unnecessary to decipher SAR for the third time. 16.  Line 166. “SA synthesis rapidly follows pathogen detection through receptors such as PRRs and NBLRRs” – this is important and more information should be provided and not here, but above. 17.  Lines 193-194. “The harms caused by drought stress to plants and crops include water deficiency, hindered transpiration, wilting and withering, nutrient imbalance, disrupted physiological metabolism, decreased yield, and increased biotic stress.” – This sentence just repeats what was said in the first sentence of the section. 18.  Line 209. “length, radicle length, root length” – I wonder how authors distinguish radicles and roots. They actually mean the same. 19.  Line 230-231. “Among the factors limiting crop production, soil salinization is a major environmental challenge. For example, according to Fu et al. phytohormones such as ABA, auxin, CK, BRs, JA, GA, SA, and ETH play certain roles in enhancing crop salt tolerance [43].” – Why the second sentence starts with “for example”? Do authors mean that hormones are major environmental challenges? 20.  Line 233. What do authors mean by “penetrability-related damages”? Such a term is absent in the literature. 21.   Line 239. “imbalance of water balance, disruption of ion balance, impaired” – “balance is repeated 3 times. Cannot it be “water imbalance, disruption of ion homeostasis”? 22.   Description of salt –stress is chaotic. Authors demonstrate poor knowledge of the theme. I advise them to read and retell reviews describing osmotic and ionic components of salinity. 23.  Line 257. “PGPR” should be deciphered and more should be said about the action of plant growth promoting rhizobacteria on plants. 24.  Line 283. “Pollutions caused by heavy metals are involved in a wide range of ions” – meaning of the phrase is unclear. This should be modified. 25.  Line 293. Again “for example” out of place here. Where is the logic? 26.  Lines 307-308. “According to Nivedha et al., SA has been used as an analgesic, anti-inflammatory, and antipyretic agent” – this was obviously written about animals and should not be mentioned in the article concerning plants. 27.  Line 309. “SA has been reported to reduce heat stress-induced ROS production and its subsequent oxi-310 dative damage”. Heat stress is out of place in the section on heavy metal stress. 28.  Line 337. Authors should explain what epigenetic regulation means. 29.  Line 373-376. “It is widely considered that SA plays vital roles in plant stress tolerance to a wide range of abiotic stresses, including low-temperature or cold stress [37-39,118]. For in-stance, SA can enhance plants’ resistance to cold, and mitigate plants’ injuries caused by low-temperature environment.” – Again, the second sentence repeats what is said in the first one. 30.  Line 379-381. “It controls ROS levels by regulating antioxidant metabolism during chilling stress. SA treatment enhances antioxidant enzymes, thereby increasing chilling tolerance.” - Again we see similar information in the first and second sentence. 31.  Line 409-410. Abbreviations APX, MDHAR, DHAR, GR, CAT, GST, GPX, POD, Gly I, and Gly II should be deciphered and it is reasonable to provide short explanation of their role. 32.  Line 4298-429. “For example, SA can regulate the plant's resistance by activating or inhibiting key components of hormone signal transduction pathways [125,128].” – It is important to mention in short which components of hormone signal transduction pathways are involved in SA-induced resistance. 33.  Line 452. PAs should be deciphered. 34.  MeHsf8-MeHSP90.9 - One more abbreviation, which should be deciphered and its function explained 35.  What SNP means? It most frequently means single nucleotide polymorphism. I think this is not appropriate here. 36.  N-methyltransferase genes – function of these genes should be explained. Why are they mentioned?

English is satisfactory, altohugh some imrovements are necessary

Author Response

Dear experts,

Thank you very much for your kind suggestions and comments!

All your suggestions and kind comments have been responded.All the paper have been completely checked out and corrected including language presentation .Please,see the attached response.

Round 2

Reviewer 1 Report

The authors answered most of the questions and comments of the reviewers. The answers satisfied me. The manuscript may be accepted for publication.

Reviewer 3 Report

Authors carefluly addressed all of my numerous comments and I am satisified

Quality of English is good enough